# The Residential Environment and Health and Well-Being of Chinese Migrant Populations: A Systematic Review

**DOI:** 10.3390/ijerph20042968

**Published:** 2023-02-08

**Authors:** Liyan Huang, Rosli Said, Hong Ching Goh, Yu Cao

**Affiliations:** 1Centre for Sustainable Urban Planning and Real Estate (SUPRE), Faculty of Built Environment, Universiti Malaya, Kuala Lumpur 50603, Malaysia; 2School of Management, University of Suzhou, Suzhou 234000, China; 3Faculty of Built Environment, Universiti Malaya, Kuala Lumpur 50603, Malaysia

**Keywords:** residential environment, health and well-being, internal migrants, housing conditions, neighbourhood environment, residential segregation

## Abstract

China’s internal migrants suffer from marginalised housing conditions, poor neighbourhood environments and residential segregation, which may have significant implications on health and well-being. Echoing recent calls for interdisciplinary research on migrant health and well-being, this study examines the associations and mechanisms of the impact of the residential environment on the health and well-being of Chinese migrants. We found that most of the relevant studies supported the “healthy migration effect”, but the phenomenon was only applicable to migrants’ self-reported physical health rather than mental health. The subjective well-being of migrants is lower than that of urban migrants. There is a debate between the effectiveness of residential environmental improvements and the ineffectiveness of residential environmental improvements in terms of the impact of the neighbourhood environment on migrants’ health and well-being. Housing conditions and the neighbourhood’s physical and social environment can enhance migrants’ health and well-being by strengthening place attachment and social cohesion, building localised social capital and gaining neighbourhood social support. Residential segregation on the neighbourhood scale affects the health outcomes of migrant populations through the mechanism of relative deprivation. Our studies build a vivid and comprehensive picture of research to understand migration, urban life and health and well-being.

## 1. Introduction

The health and well-being of migrants living in the post-migration urban environment pose a significant public health challenge [1,2,3]. The inclusion of physical, mental and social well-being in health-related Sustainable Development Goals (SDGs) has recently highlighted its significance [4]. For migrants, adapting to a new cultural setting can be a distressing and challenging experience, potentially negatively impacting their physical and mental health [5,6]. Environmental psychology studies have revealed adverse health consequences of the poor physical and social environments experienced by migrants, such as noise, air pollution, housing stress, lack of essential neighbourhood facilities and exacerbation of interpersonal relationships [7,8,9,10]. Even if their economic situation has improved, migrants vulnerably suffer from poor general health problems due to changes in the residential environment [11].

As Amin vividly depicted, “cities are polluted, unhealthy, tiring, overwhelming, confusing, alienating”. For the migrant groups, cities are “the places of low-wage work, insecurity, poor living conditions and dejected isolation” [12]. This marginalised scenario of urban life may be particularly true for China’s rural-urban migrants. They leave their familiar countryside to seek better-paid jobs and opportunities, but find themselves situated in a highly precarious urban life with socio-economic disadvantages, housing inequality, neighbourhood deprivation and hukou-based social exclusions [13,14]. Therefore, in the post-migration stage, it is critical to pay more attention to the health and well-being of Chinese migrant groups in the urban environment.

Indeed, there has been a growing body of literature on the health and well-being of migrants, as public health concerns from this increasing internal population mobility have received much attention from researchers in the urban environment [15]. Scholars have mainly studied the health and well-being of China’s migrant population in the following aspects. First, socio-economic status is an important factor, contributing to migration stress and the health and well-being of migrants. Migrant workers who experience more stress in financial- and employment-related difficulties are more likely to be mentally unhealthy [16,17]. This is due to, first, migrant workers frequently choosing low-paid “3D” jobs (dirty, dangerous and demanding), increasing their risk of exposure to hazardous and polluted environments and damaging their physical and mental health [18,19]. Secondly, the lack of hukou-based benefits is an institutional barrier to health and well-being for migrants, such as health insurance, social welfare and housing support [20,21]. A study of the health-related quality of life (HRQOL) of rural-to-urban migrants in Wuhan reported that the utilisation of health services remarkably affects the HRQOL of rural-to-urban migrants, but it is obvious that rural-to-urban migrants do not benefit from local health insurance policies [22]. Finally, only recently has the effect of Chinese migrants’ residential environment on their health-related quality of life received widespread attention [10,14,23]. Environmental psychology has addressed that perceived residential environment and neighbourhood attachment are key predictors of an individual’s quality of life [24,25]. However, a limited number of studies have focused on the residential environment and the health and well-being of migrants in China’s urban context. As such, this research is drawn from an environmental psychological perspective to understand the relationship between the residential environment and the health and well-being of Chinese internal migrants.

Interestingly, there is no consensus definition of the term “residential environment”, as it has multiple dimensions based on different research contexts. Given the diverse geographical levels of the psychological response of the residential environments to inhabitants from an environmental psychological perspective, “residential” can refer to the macro- to the micro-level [26], from the domestic living space to the neighbourhood, to the local open space, to the city or even larger geographical entities [24]. Moreover, the residential environment is more than just a place/space, but it includes the people who occupy those spaces. In addition to the home’s exact dimensions, the surrounding environment, both physical and social, as well as where it is located and where the resident performs a large proportion of their daily activities, must also be considered [27]. In this paper, the geographical unit of analysis is individual housing space and residential neighbourhoods. The residential neighbourhood can be considered as the bridge between home and city in people’s perception and actions of their residential environment [28]. As such, a review of the literature based on different dimensions of the residential environment contributes to a better understanding of the complex topic and provides insights for policymakers and those engaged in planning, designing and building inclusive neighbourhoods that promote the health and well-being of migrants.

A limited number of previous review studies have investigated how the residential environment shapes the health and well-being of China’s migrants. For example, Syed et al. [29] showed that living alone was the pre-condition for elderly migrants to feel socially isolated and have a low self-perceived quality of life and well-being. Li and Rose [14] reviewed the association of social exclusion factors and migrants’ mental health at the community and neighbourhood scale, such as perceived social status, social interaction, social participation and self-identity. Furthermore, changes in housing and neighbourhood environments due to residential mobility have also been increasingly linked to the health of migrants [30]. However, reviews on the space and place represented by the residential environment in shaping migrants’ health and well-being have received insufficient attention. With the increase in studies investigating associations between the residential environment and the health and well-being of China’s migrants in the urban context, it is timely to conduct a systematic and comprehensive review of the knowledge landscape to identify gaps, trends and future research.

To enrich the empirical study of geographic space represented by the residential environment in the study of the health and well-being of the Chinese migrant populations, this research aims to provide a systematic review of the association and mechanisms of the impact of the residential environment on the health and well-being of Chinese migrants, focusing on:Identification of the difference in health and well-being between migrants and urban residents;Examination of the association between the residential environment and migrants’ health and well-being;Exploring the underlying mechanisms for the influence of the residential environment on migrants’ health and well-being;Development of a conceptual framework on the impact of the residential environment on the health and well-being of China’s migrants, and to provide future research directions for prospective studies.

The rest of the paper is structured as follows. We start with a systematic review method, and this is followed by a review of the differences in health and well-being outcomes between migrants and urban locals. Then, we review the association and mechanisms of Chinese migrant residential environments on health and well-being. Finally, we conclude the summary to develop a conceptual framework and discuss future research directions and limitations.

## 2. Materials and Methods

The systematic literature review was conducted following the protocols of the “Preferred Reporting Items for Systematic Reviews and Meta-Analyses 2020” (PRISMA-2020), using which is possible to identify the current literature, its limitation, quality and potential, as well as give guidance to the plan and direction of the novel research [31]. Technically, PRISMA consists of the following steps: identify the eligibility criteria, describe the information sources, present the search strategy, specify the selection process, develop a data collection process, confirm data items, perform classification and categorisation, identify the risk of bias assessment and synthesis of results and discussion [32,33]. Given that a review of the current literature informs a comprehensive understanding of the effect of the residential environment on migrants’ health and well-being and analyses the limitations of and future research directions in this field, the study adopted a similar three-stage methodological approach following PRISMA [34,35].

### 2.1. Planning the Review

In this phase, we defined the scope, databases, search strategy and eligibility criteria. This study was performed to shed light on the effect of the residential environment on the health and well-being of China’s migrant populations. A systematic search was conducted to identify the relevant article from Web of Science (WoS), Scopus in English and CNKI (China Academic Journal Database) in Chinese.

This study focused on keywords related to the residential environment (housing, neighbourhood environment, community environment, physical environment, built environment, social environment, neighbourhood characteristics, dwelling environment) and health and well-being (mental health, physical health, self-report health, subjective well-being, happiness, psychological well-being, quality of life, QoL) and migrants (immigrant, migration, floating population). These keywords were used to search within the articles’ titles, abstracts and keywords. The researchers excluded articles in English that had not been peer-reviewed, while Chinese articles were included in the core journal of Peking University (PKU), the Chinese Social Science Citation Index (CSSCI) and the Chinese Science Citation Database (CSCD).

The keyword search with a specified search strategy was conducted on 11 November 2022, and 789 articles were retrieved from databases, including 649 articles in WoS, 15 in Scopus and 125 in CNKI. In this step, there was no restriction on the publication date of the selected studies. Furthermore, as shown in Table 1, the exclusion and inclusion criteria were established to efficiently reduce the number and difficulty of reviews and help to screen articles.

### 2.2. Conducing the Review

The initial search was filtered to choose articles according to the abovementioned inclusion and exclusion criteria. A total of 112 articles were removed due to their failure to meet the inclusionary criteria in primary data. Meanwhile, the present study restricted the publication date of relevant studies from January 2000 to 2022, and seven articles were reduced, as the spatial migration of Chinese migrants from rural to urban areas has surged in the last two decades. As shown in Table 2, there were 633 in total after excluding 6 duplicate articles and 31 invalid articles that did not have the full-text available online.

After that, 633 papers were manually screened based on secondary data, including examination of titles, keywords and abstracts, and then 174 articles were obtained. Furthermore, the selected 174 articles were further reviewed by the full text to assess their relevance to the aim of the study. After two rounds of precise screening, the number of articles was narrowed down to 92, including 72 articles in English and 20 in Chinese. Finally, the 92 papers were reviewed, classified and analysed against the criteria of formation categories, as shown in Table 3.

### 2.3. Reporting the Review

In this stage, the 91 articles were reviewed and analysed to present the results of the literature review. As depicted in Figure 1, the whole screening process was developed to analyse the selected articles according to some pre-defined categories [36].

## 3. Results

### 3.1. Descriptive Analysis

The above process of literature screening and selection determined 91 articles were eventually included in the systematic literature review. Figure 2 shows all of these were published in or after 2007, with a clear upward trend in publications that indicate a significant increase in health and well-being research over the last decade. Specifically, from 2007 to 2013, research on the impact of the residential environment on the health and well-being of China’s migrant populations was in its initial stages. Fewer than a dozen relevant studies were far from adequate for gaining a thorough understanding of the relationship between health and well-being and place in China [37]. With the advancement of “people-centred” urbanisation and Hukou reform, the amount of literature has been in a period of rapid growth since 2014. Furthermore, the release of the Outline of the Healthy China 2030 provided further impetus to academic research on the health and well-being of migrant populations in their places of destination. As such, the number of articles in this field steadily increased each year between 2014 and 2022.

In addition, the study sites were concentrated in Guangzhou, Shanghai, Beijing and Shenzhen, representing the most developed and populated regions of the Yangtze River Delta, the Pearl River Delta and the Beijing-Tianjin-Hebei region, respectively. Furthermore, 18 of the 91 reviewed studies were conducted in Guangzhou, followed by 15 studies conducted in Shanghai and 11 studies conducted in Beijing. Migrants tend to flow to areas and cities with higher economic concentration for more employment opportunities, higher wages and better public services [38]. Furthermore, the large cities in the central and southwestern regions have been important locations for empirical research, such as Chengdu and Wuhan (Figure 3). Meanwhile, 21 articles employed national data as a large-scale study to investigate the associations and mechanisms between migrants’ residential environment and health and well-being. Furthermore, only two of the ninety-one studies used qualitative research methods, with the rest relying on quantitative data.

### 3.2. The Difference in Health and Well-Being between Migrants and Local Residents

#### 3.2.1. Health Migrant Effect and Epidemiological Paradox

“Healthy migration effect” refers to the phenomenon of migrants who are healthier than the residents of the host society. International migration studies have found that migrants have a self-selection effect when making migration decisions. That is, healthy people are more inclined to move, so the initial health of migrants is generally better than the local population [39,40]. Researchers have also found that over time the health status of international migrants declines to a level on par with the local-born population, a phenomenon called an “epidemiological paradox” [41,42]. Meanwhile, those whose health status has deteriorated tend to be unable to permanently stay at the destination and engage in return migration to their country of origin, a phenomenon known as the “salmon bias hypothesis” [43].

The health migrant effect has been widely tested in studies related to China’s migrant populations. Fourteen of the selected studies compared the health status of rural-urban migrants and local urban residents: of these, six studies reported on the “healthy migration phenomenon” of rural migrants, in which the migrant population’s self-reported health was better than local residents [37,44,45,46,47,48]. Gu et al. [45] found that migrant workers were healthier than urban natives in terms of self-rated health status, perceived stress and chronic diseases due to self-selection effects. However, studies have also discovered that the “healthy migration phenomenon” is observed among migrants who self-reported their physical health, and does not apply to mental health [44,49]. In contrast, three reported “epidemiological paradox”, in which the self-rated health of the migrants was not significantly better than the local population [50,51,52]. A study by Li et al. [51] revealed changes in the health gap between migrants and urban residents due to health loss factors, such as income deprivation, living environment and work environment. A study in Wenzhou reported that there was an inherent health loss effect in the mobility experience of the migrant populations, meaning that the health gap between the migrants and urban residents continues to narrow over time until the health status is worse than that of urban residents [50].

In addition, five studies reported mixed results on the differences in mental health status between rural migrants and native residents [49,53,54,55,56]. Studies have well documented that migrant groups’ mental health is lower than that of the local population. Lu and Wang [55] investigated the impact of residential segregation on the mental health of migrants in large- and medium-sized cities in China, and revealed that residential segregation generates a stronger sense of relative deprivation, thus making migrant populations more psychologically stressed and more vulnerable to mental health impairment. Meanwhile, the mental health status of migrants improves as they become more settled in their host city [44]. In contrast, Wu et al. [56] examined the effect of streetscape green spaces on residents’ mental health in Guangzhou and found that the migrant groups had slightly better mental health than the local residents, possibly due to the higher visibility of green space in the urban villages where the migrants lived. Li et al. [54] also found that migrants have better mental health status than their urban counterparts, but this was much lower than rural residents in western Zhejiang province. Therefore, more comparative studies are called for to exemplify the differences in the mental health status of migrants and urban residents in different residential settings.

#### 3.2.2. The Life Course Trajectory of Migrants’ Well-Being

Previous research has shown that migrants have a lower level of subjective well-being than local residents in host cities. Eleven of the reviewed studies evaluated the differences in subjective well-being between rural-urban migrants and local urban residents. Existing studies examined the differences in subjective well-being between migrant and local residents in terms of neighbourhood interaction [57], social support [58], neighbourhood social environment [59], built environment [60] and the types of neighbourhoods [61]. A recent study from Beijing reported the significant impact of the distribution of working and living on the floating population’s sentiments, and discovered that the subjective well-being of migrants is lower than local residents [62].

Chinese elderly migrants with family-oriented migration are more passive about their residential environment than urban elderly residents. The impact of the residential environment on subjective well-being in elderly migrant groups has been extensively reported by scholars. Song et al. [63] reported that older migrants in Guangzhou have lower subjective well-being than Guangzhou-born elderly due to influencing factors, such as perceived residential environment, socio-economic status and social interaction. Similarly, Wang and Liao [64] found that housing size, community satisfaction, frequency of neighbourhood interaction and community participation significantly affect the quality of life of mobile elderly people, with elderly migrants experiencing a lower quality of life compared to non-mobile elderly people.

The new generation of migrants born after the 1980s has grown to the majority of internal labour migration. An empirical study by Liang [65], based on seven major cities in China, revealed that the new generation of migrants has lower life satisfaction and subjective well-being, which is influenced by environmental adaptation factors, such as housing conditions, community environment and city evaluation. Furthermore, variation in the demands and preferences of migrants for their residential environment across the life course results in differences in the impact of the residential environment on individual well-being. Su and Zhou [60] reported that the subjective well-being of new-generation migrants is higher than that of elderly migrants in Guangzhou, and the cleanliness of the neighbourhood public environment is more helpful for improving the subjective well-being of youth, while the harmony of neighbourhood relationships contributes to the subjective well-being at the elderly stage.

#### 3.2.3. Socio-Demographic Characteristics of Migrants and Health and Well-Being

Individual and family levels of socio-economic status have a direct impact on the residents’ health and well-being. For Chinese migrants, the differences in health and well-being caused by their socio-economic status are mirrored not only in housing stratification, but also in relative deprivation due to income status, education level and household registration systems (Hukou). Individual-level characteristics, such as income and Hukou, may not only serve as influential factors for migrants’ health and well-being, but also enable or constrain people’s choice of the residential environment [66]. Although income is frequently used as a control variable in the reviewed literature, the impact of income on the health and well-being of migrants has been well documented. A positive correlation was discovered between household income and life satisfaction [67]. Meanwhile, most academics have concurred that income has no significant effect on self-rated physical health [50,68,69]. However, the impact of income on the mental health and subjective well-being of migrants and local residents has yielded mixed results. Firstly, Chen et al. [50] reported that monthly income has a significant positive effect on mental health, whereas local residents have no significant effect in the neighbourhood-level environment. According to an empirical study based on 591 migrants in Guangzhou, income has a negative impact on migrants’ mental health, possibly because higher-income groups have less time to access neighbourhood green spaces, which is detrimental to mental health [70]. In contrast, He et al. [68] found no significant effect of monthly income on the mental health of migrants. Secondly, scholars have consistently demonstrated that individual or household income has a significant effect on the subjective well-being of migrant groups, though the positive correlation or negative correlation is mixed. Zou and Deng [71] and Liu et al. [72] revealed that income is positively associated with the SWB of migrants. However, Song et al. [63] discovered the income of elderly migrants was negatively associated with their SWB when they examined the association between the perceived residential environment, social interaction and SWB of elderly migrants in Guangzhou.

Empirically, educational attainment is often considered an important socio-demographic factor affecting the health and well-being of migrants. The SWB of migrants with higher levels of education is greater [73,74]. Moreover, He et al. [68] revealed that the promotion of mental health is more effective for university-educated migrants and higher levels of education as neighbourhood housing quality improves.

In summary, internal migration in China involves not only spatial migration from rural to urban areas, but also changes and impairments in the health and well-being of migrants. China’s dualistic household registration system, diverse housing structures and differences in individual socio-economic attributes profile a multifaceted picture of migrants’ residential environments. Consequently, more empirical studies are required to support research on the disparities in health and well-being between migrants and urban residents.

### 3.3. The Residential Environment and Migrants’ Health and Well-Being

#### 3.3.1. Housing Conditions and Migrants’ Health and Well-Being

For decades, consistent evidence has shown that various characteristics of housing affect migrants’ health and well-being [75,76,77]. Chinese migrants’ housing conditions are key elements in determining their quality of life in the host city, which can also have an impact on their health and well-being. Among the articles reviewed, 19 studies demonstrated the mechanisms and extent to which housing type, housing tenure, housing price and housing quality affect migrants’ health and well-being, as shown in Table 4.

Housing Tenure

Homeownership, as a reflection of social status, is considered to be an important factor in people’s health and well-being. In the traditional Chinese cultural and social environment, migrants make homeownership a vital goal for working and settlement in the host city. Zhu et al. [78] demonstrated that housing tenure was conducive to enhancing the social integration of the ageing pre-1970 in Beijing. Housing purchases and improving urban integration of migrants can both enhance their sense of happiness [79]. A recent study by Liu et al. [80] reported that homeownership substantially reduces the sense of relative deprivation among the migrant populations in the destination compared to the local urban residents. Furthermore, Xie [81] reported a significant impact of home ownership on the mental health of Chinese migrants and suggested that migrants who dwell in their own homes live more comfortably and have better mental health status. However, counter-intuitively, Yu [82] revealed a higher probability of self-reported illness among migrant homebuyers compared to their counterparts whose employers provided housing, possibly due to poorer physical health as a result of the greater financial stress caused by purchasing housing, but determined no significant difference in mental health.

**Table 4 ijerph-20-02968-t004:** Housing conditions, migrant’s health and well-being included in the studies.

Author, Year	HT	HS	HP	HC	HH	HF	HD	LP	HQ	HI	HL	IE
Zhou and Guo, 2022 [69]	√	√								√		
Liao et al., 2022 [74]			√									
Huang and Chen, 2022 [83]	√			√				√				
D. Liu et al., 2022 [80]					√							
Xie and Chen, 2021 [84]	√	√		√					√		√	
Zhu et al., 2022 [69]	√				√							
Chen et al., 2021 [85]			√		√			√				
You et al., 2022 [86]	√	√				√						√
Hu et al., 2020 [87]			√			√						
Zhang et al., 2019 [79]					√							
Xie, 2019 [81]		√			√				√		√	
Li et al., 2019 [88]		√					√		√			
Xiao et al., 2018 [89]	√	√				√						
J. Li and Liu, 2018 [13]		√		√		√				√		
Yu, 2016 [82]	√				√	√		√				
Yi and Zhu, 2015 [90]	√					√						
G. Wang et al., 2011 [48]	√				√	√						√
Niu et al., 2011 [91]	√				√	√		√				√
Bork et al., 2011 [92]						√						

Abbreviations: HT = housing type; HS = housing size; HP = housing price; HC = housing cost; HH = housing homeownership; HF = housing facility; HD = housing design; LP = living pattern; HQ = housing quality; HI = housing instability; HL = housing location; IE = indoor environment.

Housing type

Existing studies have found that the type of residence influences health and well-being. Zhou and Guo [69] found that housing types matter for health only among rural migrants, but not among urban locals in Chinese cities. The provision of public rental housing by the government and rental housing in relatively good living conditions increases the settlement intention of rural migrants [83]. Furthermore, there is a significant correlation between the housing type and identity expression of in situ urbanised migrants [84]. A survey of 425 migrants conducted by Yu [82] revealed a negative correlation between housing type and the mental health of new-generation migrants in Wenzhou City, Zhejiang Province; i.e., new-generation migrant workers living in buildings had better mental health. Xie [81] reported that rural migrants who lived in factory dormitories showed better mental health than those who dwelled in public rental housing or private rental housing, but there was no statistically significant difference with rural migrants who lived in self-owned housing. By comparison, two studies examined the impact of housing type on migrants’ health [48,91], and both showed no significant effect of housing type on migrants’ physical health and mental health.

Housing size and facilities

Housing size and housing facilities are important components of housing characteristics. Zhou and Guo [69] showed that housing size only has a significant impact on the health of rural migrants, as the average living space of rural migrants is much smaller than that of urban locals. Again, Xie [81] suggested that every one-square meter increase in average housing size improves rural migrants’ mental health by approximately 0.02 units. However, You et al. [86] conducted a survey of 700 migrants in Hangzhou city and reported different results, revealing no significant relationship between per capita living space and the health of migrants, and argued that the majority of migrants lived in small living spaces in the study area. Furthermore, Hu et al. [87] found that households that lived in houses with good housing facilities had a high level of overall happiness, including a kitchen, garden, balcony, internet, gas, bathroom, heating system, toilet and courtyard. Li and Liu [13] also reported that migrants who had access to better in-house facilities predicted a lower perceived stress level. Good housing facilities reduce the probability of illness and are significantly and positively associated with migrants’ self-rated health [90]. Surprisingly, a study based on migrants in Shanghai showed that housing conditions, including housing facilities, did not have a significant direct effect on migrants’ mental health, which was mediated via neighbourhood satisfaction [89].

Housing price and housing cost

High housing burdens can have a negative influence on the health and subjective well-being of migrants looking to settle in host cities. Liao et al. [74] revealed that high house prices can cause a “family split” effect and a “work more” effect, especially for less educated, female migrants and rural migrants, which can markedly affect their subjective well-being. Nevertheless, for those who are house owners, the housing value is positively correlated with their subjective well-being [87]. This is because a high-value home is not only an important way of building household wealth, but also a symbol of social status. Therefore, housing costs or housing expenditure can provide an objective picture of migrants’ housing affordability. Interestingly, a higher proportion of housing expenditure predicts a higher probability of rural migrants’ willingness to settle [83]. Even though, migrants facing high housing cost burdens tend to have a higher level of perceived stress [13].

In addition to the above, scholars have documented the effects of housing instability, living patterns, housing location and indoor air quality in influencing the health and well-being of migrants. For example, Zhou and Guo (2022) reported that housing instability has a significant impact on urban locals’ health, but only rather significantly affects the health of rural migrants. Yu (2016) found that living with family members helped new-generation migrants receive more material support and emotional comfort, and promoted their physical and mental health. Additionally, the indoor environment, including air quality, building ventilation, room lighting, noise and damp conditions, also has a potential impact on the health of migrants [82,86,90].

#### 3.3.2. Neighbourhood Environment and Migrants’ Health and Well-Being

The impact of the neighbourhood environment on residents’ health and well-being has been a key topic of interest in recent years in the multidisciplinary fields of public health, sociology, geography and urban planning [93,94,95]. For migrants, the neighbourhood is an important environment for their life and work. The quality of the neighbourhood environment not only has a direct impact on the convenience and comfort of the migrant population’s daily life [96], but is also a key place for the migrants to gain social capital and social support, which in turn affects their health and well-being [50]. Considering 27 instances in the literature, two opposing views—the effectiveness of residential environment improvement and the ineffectiveness of residential environment improvement—have evolved regarding the impact of the neighbourhood environment on migrants’ health and well-being in general [68]. Table 5 presents the associations of neighbourhood environment on migrants’ health and well-being in the reviewed 27 literatures.

Neighbourhood physical environment

Scholars who support the effectiveness of residential environment improvement have concurred that bettering the physical environments of a neighbourhood improves the living conditions, accessibility to transportation and availability of healthcare services, which consequently enhances residents’ well-being, health and cognitive abilities. Chen et al. [50] reported that the accessibility of community public services had a significant positive effect on both the self-rated health and mental health of migrants. For elderly migrants, closeness to financial facilities is negatively associated with subjective well-being, while the density of recreational facilities and density of medical facilities are positively associated with subjective well-being [97]. Furthermore, the relationship between green spaces and the mental health of migrants has been extensively studied and various pathways have been proposed. Firstly, green spaces can reduce environmental stressors such as air pollution and noise, both of which are thought to be detrimental to migrants’ mental health [98]. Secondly, green landscapes, as a health promotion resource, help to restore concentration, reduce psychological stress and awaken positive emotions, increasing the life satisfaction of residents [70]. It has also been posited that open spaces increase the opportunities to facilitate physical exercise and effectively promote physical and mental health [67]. Finally, green space can enhance place attachment and social cohesion, which is protective of migrants’ mental health. For instance, Wu et al. [56] studied the impact of streetscape green spaces on residents’ mental health in Guangzhou, and suggested that streetscape green spaces increased residents’ place attachment and led to improvement in the mental health of migrants. Yang et al. [67] suggested that actively utilizing green spaces improves social cohesion by increasing social interaction and fostering social connections in Shenzhen. Meanwhile, several studies have observed the positive effects of neighbourhood cleanliness, aesthetic quality and architectural amenities on the health and well-being of migrants [59,99].

**Table 5 ijerph-20-02968-t005:** Neighbourhood environment, migrants’ health and well-being included in the studies.

Author, Year	Study Design	Physical Environment												Social Environment								
		RF	FF	HF	CF	GS	TF	PS	AQ	NC	PF	BD	NL	SC	SI	NS	SS	NT	SP	SN	ST	RD
Y. Liu et al., 2022 [100]	QN	√	√	√										√								
Song et al., 2022 [63]	QN														√	√						
Z. Zhang et al., 2022 [98]	QN					√									√							
Y. Liu, et al., 2022 [97]	QN			√		√	√							√								
Yang et al., 2022 [101]	QN					√																
Yang et al., 2022 [102]	QN															√	√					
Y. Wang and Lo, 2022 [103]	MM					√					√											
K. Zhang et al., 2022 [104]	QN						√			√				√		√						
M. Yang et al., 2021 [66]	QN	√	√	√		√			√					√		√						
Pan et al., 2021 [105]	QN	√	√	√		√	√									√						
Tian et al., 2021 [106]	QN	√														√				√	√	
Su and Zhou, 2021 [60]	QN	√		√			√			√	√				√	√						
X. Chen et al., 2021 [50]	QN	√			√		√			√	√		√	√		√						
Wu et al., 2021 [56]	MM					√								√								
Lei and Lin, 2021 [107]	QN													√				√				
Zou and Deng, 2021 [71]	QN																	√		√		
He et al., 2021 [68]	QN				√		√								√			√				
M. Yang et al., 2020 [70]	QN					√								√								
Qiu et al., 2019 [99]	QN					√						√			√	√						
J. Li et al., 2019 [88]	QN	√		√	√						√					√						
Y. Liu et al., 2019 [57]	QN														√							√
S. Lin and Huang, 2018 [108]	QN					√	√				√											
Y. Liu et al., 2018 [59]	QN						√			√							√					√
Y. Liu et al., 2017 [58]	QN									√	√						√					√
Gu et al., 2015 [45]	QN									√	√			√		√	√					
Bork et al., 2011 [92]	QN									√	√											
Wen et al., 2010 [37]	QN													√		√						

Abbreviations: RF = Recreation and fitness facilities; HF = Health facilities; CF = Commercial facilities; FF = Financial facilities; PF = Public facilities; GS = Green space; PS = public space; AQ = Aesthetic quality; TF = Transportation facility; NC = Neighbourhood Cleanliness; BD = Building Density; NL = Neighbourhood Location; SI = social interaction; SC = social cohesion; SS = social support; NS = neighbourhood Safe; NT = neighbourhood type; SP = social participation; SN = social network; ST = social trust; RD = Relative Deprivation; QN = quantitative method; MM = mixed method.

However, scholars who contend the ineffectiveness of residential environment improvement have suggested mixed results. A study by Yang et al. [109] on residents’ self-rated physical and mental health in terms of the built environment and community integration factors in Beijing showed that built environment indicators had a relatively weak effect on residents’ physical and mental health. Following this, Yang et al. [67] further demonstrated that the direct effect of the built environment on life satisfaction was weaker in Beijing, with neighbourhood social interaction and neighbourhood satisfaction having a greater impact on residents’ health and well-being than the objective environment. Furthermore, Yang et al. [70] reported that no direct effect of perceived green space on migrants’ mental health was found, but rather enhanced migrants’ mental health by reducing perceived environmental disturbance and increasing social cohesion. As such, studies on the impact of the physical community environment on migrants’ health and well-being have shown a complex and contradictory picture that needs to be tested and supported by further empirical research.

Neighbourhood social environment

The neighbourhood social environment is defined as the sociodemographic composition of the neighbourhood and its residents, as well as the relationships, groups and social processes that exist among individuals living in the neighbourhood [110]. There are significant differences in the socio-demographic characteristics of residents living in different settlements, which in turn contribute to differences in health and well-being. For example, He et al. [91] found that the higher the proportion of locals in the neighbourhood, the better the self-rated health status and mental health of migrants due to the neighbourhood effect. In addition, ten of the reviewed articles consistently revealed that social cohesion is positively associated with migrants’ health and well-being [37,45,50,56,66,70,97,100,104,107]. Good social cohesion is directly linked to a highly subjective sense of well-being among older migrants [97,100]. Yang et al. [66] conducted a survey of 591 migrants in Shenzhen and found that a perceived decline in social cohesion is associated with poor mental health. In contrast, Chen et al. [50] observed no significant effect of social cohesion on the self-rated health and mental health of migrants, and suggested that the effect of social cohesion on the health of local urban residents was significantly greater than that of the migrants.

Studies in Western countries have shown that neighbourhood interaction not only enhances residents’ overall evaluation of their state of life but also contributes to the control of negative emotions and the development of positive emotions [111,112]. There is also a growing body of research on the impact of social interaction on the health and well-being of Chinese migrants. Liu et al. [57] examined the buffering effect that neighbourhood interaction had on environmental stresses of community poverty and contributed to improving the subjective well-being of migrant populations. Social interaction with friends and neighbours, as well as mutual neighbourhood support, have a substantial impact on the subjective well-being and quality of life of older migrants [62,105].

Neighbourhood safety is important for residents’ health and well-being [113,114]. Yang et al. [66] revealed that the safety of the living environment was more strongly associated with reduced levels of psychological stress among migrants than any other physical neighbourhood characteristics. However, Wen et al. [37] argued that neighbourhood safety and social cohesion are health-promoting resources exclusively for natives rather than migrants in Shanghai, because these so-called “floating populations” may not have been sufficiently exposed to their neighbourhood environment to manifest a neighbourhood effect. Similarly, Chen et al. [50] also observed that improvements in neighbourhood safety were not related to the mental health benefits of migrants, but only to their self-rated health status.

#### 3.3.3. Residential Segregation and Migrants’ Health and Well-Being

Residential segregation refers to the degree of difference in the composition and spatial distribution of the population with different social characteristics across its neighbourhoods in an entire metropolitan area [115]. Residential segregation shapes the unique neighbourhood environment of segregated communities, which is transmitted to individuals through neighbourhood effects and, in turn, has an impact on their health and well-being [116]. Lu and Wang [55] reported that the more segregated the neighbourhoods, the poorer the mental health of the migrants, and that residential segregation is more detrimental to the mental health of urban-to-urban migrants than rural-to-urban migrants. Two studies examining the impact of residential quality on the health status of migrants also consistently demonstrated that residential segregation had a significant impact on both self-rated health and psychological health, and the impact of residential segregation on the health of migrants significantly decreased as the duration of stay increased [90,117].

### 3.4. Influence Mechanisms of the Residential Environment on the Migrants’ Health and Well-Being

#### 3.4.1. The Mediating Effect

The residential environment of the host city can contribute to the health and well-being of migrants not only directly, but also directly or indirectly through place attachment, social cohesion, social capital, neighbourhood effect and social support. Wu et al. [56] determined that place attachment plays a partially mediating role in the mental health of migrants in Guangzhou; i.e., streetscape green spaces can directly enhance the place attachment of migrant residents, which in turn indirectly enhances the psychological well-being of migrants. Using a sample of 591 migrants in Shenzhen, Yang et al. [70] revealed that social cohesion mediates the impact of residential green space on the mental health of internal migrants, meaning that migrants’ perceived green space improves their mental health by promoting social cohesion. Furthermore, several studies consistently validated the mediating effect of social capital in the residential environment on migrants’ health and well-being. For example, Song et al. [64] suggested that social interactions mediated the association between perceived residential surroundings and self-reported subjective well-being among older migrants, indicating that social interactions with friends and neighbours have a significant impact on elderly migrants’ subjective well-being. Zou and Deng [118] discovered that neighbourhood choice did not directly affect the socio-economic integration of Chinese migrants, partly due to the building of localised social capital, which increased migrants’ willingness to settle and their level of social integration. There is a “surprising” negative correlation between neighbourhood socioeconomic status and neighbourhood social capital, as old neighbourhoods create the conditions for the accumulation of social capital and contribute to the health of elderly migrants [107]. Perceived social capital can alleviate the relative deprivation of housing attributes for migrants [80], and linking social capital also plays a significant mediating role between subjective relative deprivation and life satisfaction [119].

In addition, He et al. [68] explained the neighbourhood effect as an explanatory mechanism for the impact of community economic characteristics and neighbourhood composition on the differences in the health of migrants at the community level. Residential segregation in the neighbourhood affects the mental health of migrant populations through mechanisms of relative deprivation [55]. Housing conditions have indirect effects on migrants’ mental health via the mediating role of neighbourhood satisfaction [89]. The perceived living standards relative to multiple reference groups play the mediating role of household income and home ownership in the host city on first- and second-generation migrants’ subjective well-being [72].

#### 3.4.2. The Moderating Effect

Exploring moderating effects provides a research snapshot of a comprehensive understanding of the underlying mechanisms of the impact of the residential environment on migrants’ health and well-being. A survey of 712 residents from Guangzhou showed that residential mobility moderates the association between neighbourhood environment and subjective well-being, with the relationship between perceived neighbourhood environment and well-being strengthening as the length of mobility increases [60]. Liu et al. [88] suggested neighbourhood interaction can buffer the negative effects of neighbourhood poverty on migrants’ well-being, despite no significant moderating effect of social interaction on neighbourhood deprivation on subjective well-being being observed. Moreover, Cheng et al. [73] conducted a survey of 2573 migrants in nine metropolitan areas in China, and observed that gender factors had a moderating effect on the impact of residential instability on their self-rated health and subjective well-being.

## 4. Discussion

### 4.1. Summary of Findings and Discussion

The massive migrant populations in China moving from rural to urban areas provides a vivid and expansive social scenario for examining the association between the residential environment and health and well-being. Internal migration in China is embedded in the rural-urban dual structure of housing policy, Hukou system and the social welfare system, which has a crucial impact on the residential environment and the health and well-being of migrants [120]. This review has synthesised current empirical research on the impact of the residential environment on the health and well-being of migrant populations in China, comparing the differences in health and well-being between migrants and urban residents, examining the associations and underlying mechanisms of influence between residential environment on health outcomes of migrants, and attempting to develop a conceptual framework for studying the influence of the residential environment on the health and well-being of migrants in China. The continuing urbanisation of developing countries represented by China and the vulnerability of internal migrant groups to health and well-being in destination cities demonstrate the importance of research on this topic. However, a limited number of studies have focused on the residential environment and the health and well-being of migrants in China’s urban context, and less attention has been paid to reviews on the residential environment represented by place/space and multiple-layered residents in shaping migrants’ health and well-being.

This review focuses on the association between the residential environment and migrants’ health and well-being, with a particular emphasis on the residential environment as a concept that integrates aspects of housing conditions, the physical and social environment of existing studies on neighbourhoods and residential segregation. The results of our review of the available studies partially echoed the findings reported by Li and Rose [14], as the latter conducted a systematic review of studies focusing on the mental health of migrants. We found that most of the relevant studies supported the “healthy migration effect”, but the phenomenon was only applicable to migrants’ self-reported physical health rather than mental health. The finding that migrants have a generally lower level of mental health than urban natives is consistent with the findings by Li and Rose [14]. Moreover, the available studies consistently demonstrated that the subjective well-being of migrants is lower than that of urban migrants. Furthermore, from a life course perspective, there are significant differences in the health and well-being of elderly migrants and young migrants. The extent to which housing and the neighbourhood environment influence health outcomes significantly differs between men and women. Furthermore, income, education level, family structure and work intensity are identified as potential influences related to migrants’ health and well-being in the reviewed papers.

Previous research on the association between residential environment and migrants’ health and well-being presents a complex and contradictory picture. Most of the studies we were able to include in our review only focused on one aspect of the residential environment, such as housing characteristics, neighbourhood environment or residential segregation. In addition, there are several research controversies in these reviewed studies, such as the debate between the effectiveness of residential environment improvement and the ineffectiveness of residential environment improvement [68]; whether environmental factors have a weaker impact on migrants’ subjective well-being than individual socio-economic attributes [100]. Therefore, further research is needed to test the association between the residential environment and migrants’ health and well-being.

Analysis of the influence mechanisms of the residential environment on the health and well-being of migrants is key to establishing a causal relationship. In short, housing conditions and the neighbourhood’s physical and social environment can enhance migrants’ health and well-being by strengthening place attachment and social cohesion, building localised social capital, and gaining neighbourhood and community social support. Residential segregation on the neighbourhood scale affects the health outcomes of migrant populations through the mechanism of relative deprivation. In addition, the frequency of residential migration, level of neighbourhood interaction and gender were identified as moderating variables for differences in the impact of the residential environment on migrants’ health and well-being. As such, based on a systematic review of the reviewed literature, we constructed a conceptual framework on the impact of the residential environment on migrants’ health and well-being (Figure 4).

### 4.2. Future Research Recommendations

In future research, further investigations on the relationship between the residential environment and migrants’ health and well-being should take into account the heterogeneity of Chinese migrant groups, such as the different stages of life courses, gender differences and the change in migration pattern. Firstly, less attention has been paid to the issue of adaption to the new residential environment and the potentially important role in migrants’ health and well-being at different stages of life courses [60]. The change in migrants’ residential environment at their different stages of life courses and its impact on their health and well-being requires further study. Moreover, very few studies have revealed that gender differences exist in the degree to which residential mobility affects the physical health and subjective well-being of migrants [73]. It is necessary to respond to the impact of gender differences on the relationship between the residential environment and the health and well-being of migrants in the context of fertility policy and tradition culture. Furthermore, fewer studies have focused on tracking changes in the residential environment of migrants from individual mobility to family migration, which in turn leads to differences in their health and well-being. As such, future research needs to fill these research gaps and provide more empirical evidence for research on the health and well-being of Chinese internal migrants.

Additionally, a handful of studies have examined micro-scale spatial variation in the health and well-being of the migrant populations and revealed the interactive effects of housing status and neighbourhood environment [68,86]. This is surprising given that existing research remains doubtful regarding the effectiveness of residential improvements in enhancing the health and well-being of migrants. More empirical research is therefore required to respond to this debate. Furthermore, it is worth noting that few studies have examined the effect of residential segregation on the health and well-being of migrants [55,117]. The neighbourhood environment shaped by residential segregation transmits environmental characteristics to individuals and has a significant impact on their QoL [121]. Consequently, future research could attempt to investigate the impact of the residential environment, including housing conditions, the built and physical environment of the neighbourhood and residential segregation, as well as the socio-economic characteristics of individuals, on migrants’ health and well-being, to comprehensively explore the extent to which improvements in the residential environment affect the QoL of migrants.

Although the literature concerning the effect of the residential environment on QoL is accumulating, there are far fewer empirical studies examining the underlying mechanisms of the residential environment on migrants’ health and well-being to establish causal relationships. Moreover, previous studies have focused on the mediating or moderating effects of individual environment perceptions and social contact, and have underexplored the potential mechanisms of influence from cultural factors. The effect of acculturation and ethnic identity on the health and well-being of immigrants dwelling in different settlements in developed countries has been widely tested [122]. However, how acculturation and identity affect the health and well-being of migrants in different neighbourhood environments has not been studied in depth in China. Furthermore, there is still debate regarding the degree of social integration of migrants in terms of cultural adaptation, identity and psychological integration in the different types of neighbourhoods in China, which has a mixed impact on their health and well-being [123]. Therefore, further research on how social integration plays a causal mechanism in the impact of the residential environment on the health and well-being of migrants is needed.

Last but not least, existing studies have been mainly cross-sectional, with few longitudinal studies conducted to examine changes in residential mobility and the health and well-being status of migrants at different stages of migration. As a result, we emphasise the importance of longitudinal studies comparing changes in living environments and differences in the health and well-being status of migrants as a result of residential mobility across the life course, as well as the establishment of more tracking surveys based on the health and well-being of individual migrants living in cities.

### 4.3. Limitations

This review study is limited in the literature retrieved from the WoS core collection, Scopus as well as the Chinese core journal database, so our results may be slightly different if we were to refer to other literature sources, such as Google Scholar or Chinese non-core journals. Secondly, the results reviewed mostly focused on cross-sectional studies, with few studies longitudinally comparing the residential mobility of migrants and changes in health and well-being at different stages of migration. Finally, the systematic review was based on qualitative analysis and may be at risk of selection bias, which means that the results of this study must be interpreted with caution.

## 5. Conclusions

The health and well-being of migrants are an important concern for migration studies worldwide. The mass spatial transformation of migrants makes them vulnerable to living in segregated and poorly maintained areas, as well as exposure to exclusion from basic services, which has a negative impact on the health and well-being of migrants. This study presented a systematic literature review revealing the associations and mechanisms of the effects of the residential environment on the health and well-being of Chinese migrants in destination cities. The findings suggested that housing conditions, the built and social environment of the neighbourhood and residential segregation have significant and complex associations and multiple pathways of mechanisms on migrants’ health and well-being. Although our study is limited to one country, our research further strengthens the link between the impact of the residential environment on the health and well-being of migrant groups.

## Figures and Tables

**Figure 1 ijerph-20-02968-f001:**
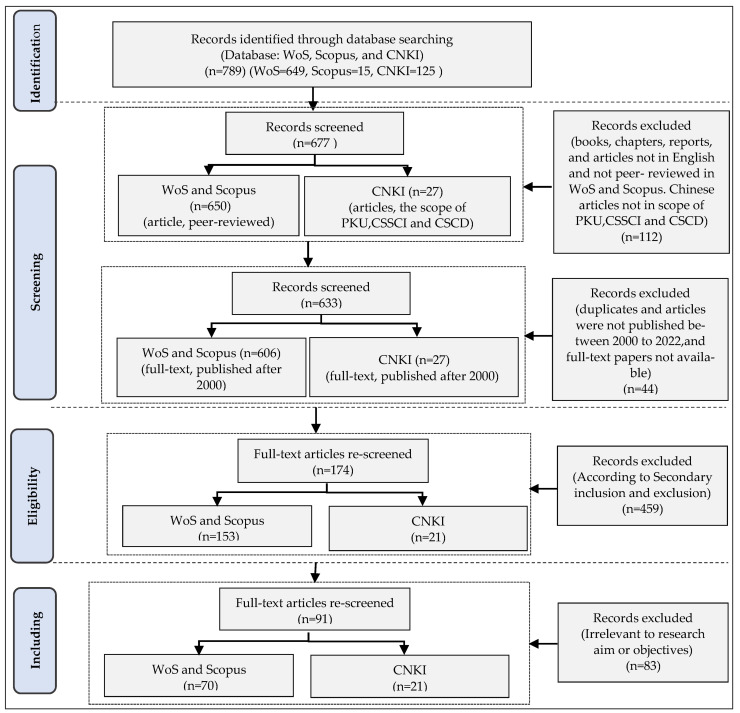
Flow diagram of article selection process.

**Figure 2 ijerph-20-02968-f002:**
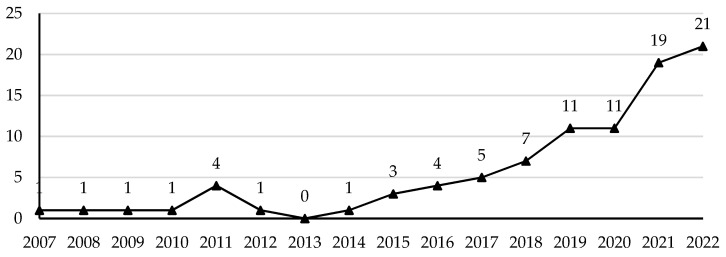
The number of selected articles published per year.

**Figure 3 ijerph-20-02968-f003:**
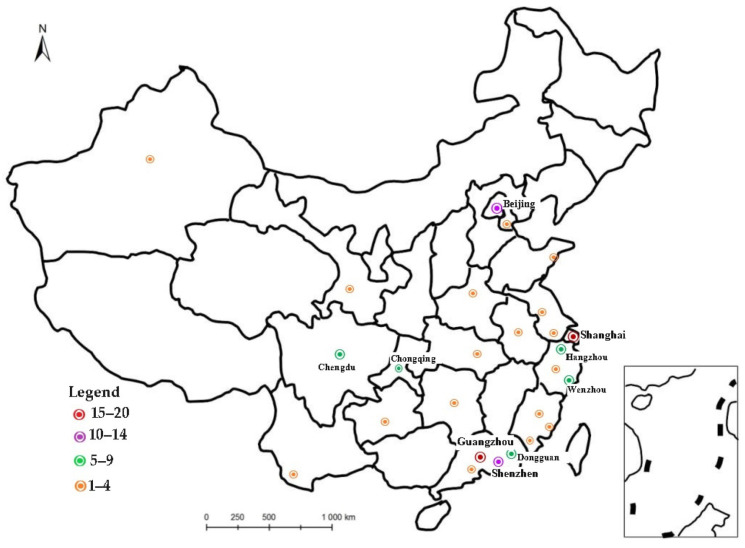
Distribution of the articles by city.

**Figure 4 ijerph-20-02968-f004:**
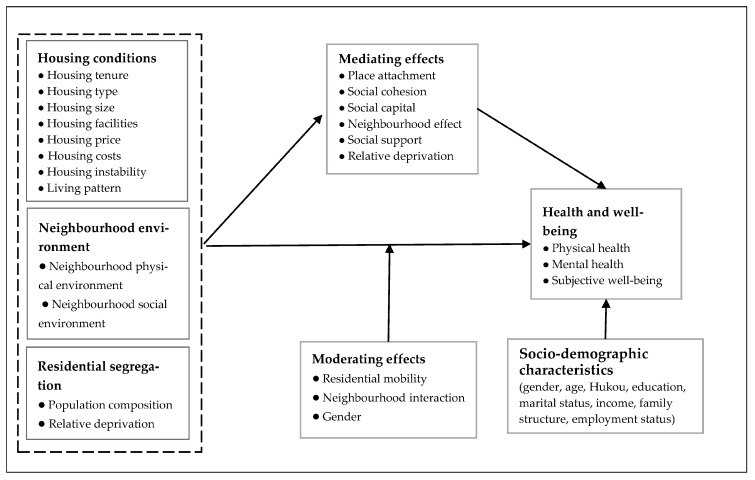
The conceptual framework of the impact of residential environment on migrants’ health and well-being.

**Table 1 ijerph-20-02968-t001:** Inclusion and exclusion criteria.

Primary Data		Secondary Data	
Inclusionary	Exclusionary	Inclusionary	Exclusionary
Journal articles in WoS, Scopus and CNKI	Duplicate records	The residential environment in shaping the health, well-being and quality of life of China’s migrants	Not related to the residential environment and in shaping the health, well-being and quality of life of China’s migrants
Conference paper in WoS, Scopus and CNKI	Books and chapters	Relevant to the research objective	Irrelevant to the research objective
Peer-reviewed in English	Invalid articles or papers		
The scope of PKU, CSSCI and CSCD in Chinese			
Full-text available online			

**Table 2 ijerph-20-02968-t002:** The search string and results of the WoS, Scopus and CNKI databases.

Scientific Database	Search String	Results
WoS	TS = ((residential environment OR housing OR living conditions OR neighbourhood environment OR community environment OR physical environment OR social environment OR built environment OR neighbourhood characteristics OR dwelling environment) AND (health OR well-being OR “mental health” OR “physical health” OR “self-report health” OR “subjective well-being” OR happiness OR “psychological well-being” OR “quality of life” OR QoL) AND (migra* OR immigra* OR “floating population”) AND (China OR Chinese))	649
	Document Types: Articles or Proceedings Papers or Review Articles	647
	Languages: English	640
	AND Publication years: 2000–2022	634
Scopus	TITLE-ABA-KEY ((residential AND environment OR housing OR living AND conditions OR neighbourhood AND environment OR community AND environment OR physical AND environment OR social AND environment OR built AND environment OR neighbourhood AND characteristics OR dwelling AND environment) AND (health OR well-being OR “mental health” OR “physical health” OR “self-report health” OR “subjective well-being” OR happiness OR “psychological well-being” OR “quality of life” OR QoL) AND (migra* OR immigra* OR “floating population”) AND (China OR Chinese))	15
	AND (LIMIT-TO (DOCTYPE, “ar”) OR LIMIT-TO (DOCTYPE, “re”)) AND (LIMIT-TO (LANGUAGE, “English”))	10
	AND Publication years: 2000–2022	9
CNKI	TS = (residential environment + neighbourhood environment + living conditions + housing conditions + built environment) * (health + well-being + mental health + quality of life) * (floating population + migrant)	125
	Document Types: Journal	55
	Source category: PKU, CSSCI and CSCD	27
	AND Publication years: 2000–2022	27

Note: * indicates any group of characters, including null characters.

**Table 3 ijerph-20-02968-t003:** Selection criteria for formulation categories.

Step	Selection Criteria
1	Identify the relevant articles focusing the RE on the health and well-being of Chinese migrants.
2	Determine the difference in RE and the health and well-being between migrants and urban residents.
3	Categorize the effect of RE on Chinese migrants’ mental health consequences.
4	Group the potential pathways for the influence of RE on migrants’ health and well-being.
5	Review the selected literature again and update the shortlisted categories.
6	The final categories are verified, classified and finalized.
7	Relevant categories are distributed and selected under the most pertinent categories.

## Data Availability

Not applicable.

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
