# Peer review of "The Residential Environment and Health and Well-Being of Chinese Migrant Populations: A Systematic Review"

_ijerph, 2023, doi:10.3390/ijerph20042968_

Round 1

Reviewer 1 Report

The study is well planned and articulated. While the research is based on work that is done, the authors tried to provide a unique perspective to the body of work on the effects of migration in China. The effects of migration is obvious, especially rural to urban settings. 

There are few other factors that this reviewer would be interested in seeing the authors look at for future study: gender difference to migration, ethnicity and cultures (dialect, religion, etc). These differences may provide a deeper look at the heterogenous population described by the authors. The current study does not provide the complex make-up of the migrants. From what is written, the migrants all seem to be either rural or urban/local or stranger. This would make the study much more interesting and unique to the materials found. If the previous studies did provide this information, it would be good to follow-up on it also. No country is just one culture, ethnicity or gender. The authors may add how rural and urban people view one another. There are probably prejudices that each which may also contribute to the mental health of the migrants. 

The residential environment is more than just a place/space but it includes the people who occupy those spaces. And, people are complex entities who are multi-layered. 

Reviewer 2 Report

This is a comprehensive reviewing paper about the health and well-being of Chinese migrants. The adopted methodology and the procedure to collect the related papers were clearly summarized, which is enough for an academic paper. Some of potential factors that can be related to the health of people or QOL of migrants should be commented to improve the  level of discussions. At least, some of adequate defense to those comment is required. If possible, minor additional discussion is expected to be added.

Comment 1 for the paper searching keywords in table 2, page4 to 5; Why not the economic reason or income status of the migrants was included? Economic reason or economic demand for their working can be considered as fundamental background of migration. Also, the life-stage of migrants are not discussed. Migration can be occurred at some stages of life-stage driven  by different reasons or motivation. The case that a migrant is single and has no responsibility to support the other member of household member, or the case that a migrant is retired or the case that a migrant is obedient to the parents decision of migration will have also different impact on the migrant's health or well-beings.

Comment 2 for chapter 3; The discussion in the comment 1 is related to the design how to get conclusion from the many references. The conclusion part of such the comprehensive review tends to be derived by the number of papers with some common tendency among much number of references, and the number of papers for each sub-topics is also commented in this paper.  As well-known, Simpson's paradox can be occurred without  adequate data-stratification(s). For example, the effect of neighborhood environment is completely opposite between the low income migrants and high income migrants, the conclusion obtained from the set of papers with no stratification is not consistent with that with an adequate stratification.  Please add some comments or discussion on the paper or make a defense to this comment.
